# Genome-Wide Association Analysis for Tuber Dry Matter and Oxidative Browning in Water Yam (*Dioscorea alata* L.)

**DOI:** 10.3390/plants9080969

**Published:** 2020-07-31

**Authors:** Cobes Gatarira, Paterne Agre, Ryo Matsumoto, Alex Edemodu, Victor Adetimirin, Ranjana Bhattacharjee, Robert Asiedu, Asrat Asfaw

**Affiliations:** 1International Institute of Tropical Agriculture (IITA), Ibadan 200001, Nigeria; P.Gatarira@cgiar.org (C.G.); R.Matsumoto@cgiar.org (R.M.); A.Edemodu@cgiar.org (A.E.); r.bhattacharjee@cgiar.org (R.B.); r.asiedu@cgiar.org (R.A.); a.amele@cgiar.org (A.A.); 2Pan African University, Institute of Life and Earth Sciences, University of Ibadan, Ibadan 200001, Nigeria; 3Department of Agronomy, University of Ibadan, Ibadan 200001, Nigeria; votimirin@yahoo.com

**Keywords:** DArT sequencing, gene annotation, marker–trait association, yam tuber quality

## Abstract

Yam (*Dioscorea* spp.) is a nutritional and medicinal staple tuber crop grown in the tropics and sub-tropics. Among the food yam species, water yam (*Dioscorea alata* L.*)* is the most widely distributed and cultivated species worldwide. Tuber dry matter content (DMC) and oxidative browning (OxB) are important quality attributes that determine cultivar acceptability in water yam. This study used a single nucleotide polymorphism (SNP) assay from a diversity arrays technology (DArT) platform for a genome-wide association study (GWAS) of the two quality traits in a panel of 100 water yam clones grown in three environments. The marker–trait association analysis identified significant SNPs associated with tuber DMC on chromosomes 6 and 19 and with OxB on chromosome 5. The significant SNPs cumulatively explained 45.87 and 12.74% of the total phenotypic variation for the tuber DMC and OxB, respectively. Gene annotation for the significant SNP loci identified important genes associated in the process of the proteolytic modification of carbohydrates in the dry matter accumulation pathway as well as fatty acid β-oxidation in peroxisome for enzymatic oxidation. Additional putative genes were also identified in the peak SNP sites for both tuber dry matter and enzymatic oxidation with unknown functions. The results of this study provide valuable insight for further dissection of the genetic architecture of tuber dry matter and enzymatic oxidation in water yam. They also highlight SNP variants and genes useful for genomics-informed selection decisions in the breeding process for improving food quality traits in water yam.

## 1. Introduction

Yams are herbaceous perennial vine plants in the genus *Dioscorea*, comprising over 600 species [1,2]. They grow in the tropics and sub-tropics of Africa, the Caribbean, Latin America, Asia, and Oceania as a source of dietary food and ingredients for pharmaceuticals and traditional medicine. In West Africa, where over 92% of its global production occurs [3], yam is involved in many key life ceremonies [4]. *Dioscorea alata*, also known by the common names ‘water yam’ or ‘greater yam’, is one of the most important food yams. It is the world’s most widely distributed cultivated yam, though it is not cultivated on the same magnitude as *D. rotundata* due to traditional bias that overlooks its nutritional and agronomic potential [5].

End-use quality significantly influences the acceptance of yam varieties by farmers and consumers [6]. In effect, the success of newly developed yam varieties depends not only on their agronomic attributes but also on their acceptability to consumers in terms of both sensory and utilization characteristics [7]. The yam tuber has diverse uses as a result of the wide variation in organoleptic, culinary and nutritional properties, making some yam cultivars more appropriate for certain types of food preparation than others. Water yam cultivars with good eating qualities are characterized by high dry matter, starch, and amylose contents [8]. The rapid change in colour of the tissue from white to yellow or brown after the tuber is cut, which is the result of the oxidation of polyphenols, influences its processing and utilization [9]. Polyphenolic oxidation is also linked to bitterness, off-flavours and in some instances necessitates special preparation processes to make acceptable dishes [10].

Boiled yam, pounded yam (also known as “fufu”), and “amala” (prepared from cooking and stirring of the fermented yam flour—“elubo”) are the three leading food forms of yams in West and Central African regions [11,12]. In this region, *Dioscorea rotundata* (white yam) is preferred to water yam for “fufu” and “amala” due to its ease of dough formation when pounded. However, some collections and advanced breeding lines of water yam at the International Institute of Tropical Agriculture (IITA, Ibadan, Nigeria) have displayed the capacity to form good dough, similar to or even superior to that of some genotypes of *D. rotundata* [7]. Additionally, some water yam clones have higher total dietary fibre and amylose content than that reported for brown rice and whole wheat flour together with low sodium but high potassium content, indicating the nutritional role that water yam could play in managing chronic diseases including diabetes [13].

This highlights the potential of water yam for food and nutritional security in West and Central Africa, particularly as it has a very wide adaptation, high genetic potential to produce fairly high yield under low to average soil fertility, early vigour for weed suppression, ease of propagation through the production of bulbils resulting in a high multiplication ratio, low post-harvest losses, good processing quality and high nutritive value [5,14].

An improvement in the food quality of *D. alata* remains a key objective in yam breeding programmes [7] and is critical for increasing the adoption level of newly developed varieties. To date, the genetic improvement efforts to develop water yam varieties with enhanced tuber quality attributes are mainly by conventional breeding strategies based on phenotypic records/data. This is, however, arduous and slow due to the lengthy screening process of identifying superior individuals from clonal populations [5,15]. In addition, the genetic basis of traits that define tuber quality has not received much attention. With the advent of molecular markers, tremendous progress has been made to understand the genetic diversity and relationships in *Dioscorea* species including *D. alata* (water yam) [16]. The use of genotypic and phenotypic data has also proven to elucidate genetic diversity in different *Dioscorea* spp. [17].

The advancements in next-generation sequencing technologies have led to the rapid development of DNA-informed breeding techniques such as marker-assisted breeding and genomic selection through which many crops have recorded fast genetic gains. The increasing availability of molecular markers enables researchers to tag regions of the genome associated with specific phenotypes of interest in Quantitative trait locus (QTL) mapping and genome-wide association studies [18]. The genetic mapping of loci underlying important tuber quality traits of water yam has not been conducted yet to assist selection decisions in the breeding process. Mapping based on genome-wide associations has become increasingly popular and powerful because of the emergence of more cost-effective, high-throughput genotyping platforms. Using molecular approaches, such as the candidate gene technique, to unravel the causal gene(s) would hasten efforts in introgressing tuber quality traits into preferred genetic backgrounds of *D. alata*. The precision and speed of crop breeding have been improved by the evolution of genetic linkage and association mapping of the quantitative traits. This has been clearly demonstrated by Pétro et al. [19] in water yam. Genomic resources for *D. alata* are being rapidly developed [20]. These include the recent pre-release of a chromosome-scale “v2” assembly of *Dioscorea alata* see release notes and assembly [20]. This genome sequence enables genome-wide association studies of key traits in water yam.

To elucidate the genetic factors for tuber dry matter content and oxidative browning, a diversity panel consisting of 100 water yam clones was genotyped by diversity arrays technology sequencing. While whole-genome sequencing provides the highest resolution, it still remains expensive for non-model species such as yams. A genome-wide association study was conducted to identify single nucleotide polymorphism loci or QTL regions and genes associated with tuber dry matter and oxidative browning. The SNP loci and associated candidate genes, when validated, would be a valuable resource for marker-assisted selection in the breeding process to develop new water yam varieties with acceptable end-user qualities.

## 2. Results

### 2.1. Genotypic Variability for Tuber DMC and OxB

We analysed phenotypic data from the three locations to estimate the variance for total genotypic value and genotype × location effect (Table 1). The variance estimates for the two traits were statistically significant (*p* < 0.05) for total genotypic and genotype × location effects. The variance estimate for the total genotypic value was higher than that of the genotype × location effects. Genotypic differences among the diversity panel for DMC and OxB were statistically significant (*p* < 0.05) in each of the three locations (Figure 1). Mean DMC was highest in Ubiaja (32.43%), followed by Ikenne (29.20%) and Ibadan (28.83%), while mean oxidative browning was high at Ikenne, followed by Ibadan and Ubiaja. Significant negative correlation (r^2^ = −0.39, *p* < 6.825074 × 10^−5^) was obtained between DMC and the OxB.

### 2.2. Genotyping, Population Structure, and Linkage Disequilibrium

Table 2 presents the number of SNPs on *D. alata* chromosomes before and after filtering for missing data, allele frequency, and heterozygosity (see Materials and Methods for details). The SNP calling pipeline yielded 22,140 highly polymorphic SNP markers, of which 18,067 were mapped onto the 20 *D. alata* chromosomes while 4073 were unmapped. Out of the mapped markers, 9687 SNPs that qualified the filtering criteria were used as input for GWAS analysis. The filtered SNPs were not proportionally distributed across the 20 chromosomes with highest counts of 923, 893, and 843 displayed on chromosomes 5, 4, and 19, respectively (Table 2, Appendix A). The lowest number of SNPs, 299 and 304, were mapped on chromosome 1 and 13, respectively (Table 2). Minor allele frequency (MAF) across 9687 SNP markers varied from 0.05 to 0.50 with an average of 0.209. Observed and expected heterozygosity varied from 0.01 to 0.71 and 0.009 to 0.5, respectively. Across the chromosome, the polymorphism information content (PIC) of filtered SNPs varied from 0.20777 (chromosome 18) to 0.27728 (chromosome 10) with mean 0.23553 (Table 2). Genetic diversity conducted previously by Agre et al. [21] through discriminant analysis of principal components (DAPC) and admixture (population structure) revealed three major clusters with a low genetic pair-wise fixation index among clones of each group. Through admixture analysis, very few clones were identified as pure while 66% of the total accessions were found to be admixed.

Linkage disequilibrium analysis revealed 312,479 loci pairs within a physical distance that extends up to 998,066 bp. About, 6.04% (450) of the loci pairs were in significant linkage disequilibrium (LD) (*p* < 0.001). In addition, 221 (3.10%) of the pairs were in complete LD (R^2^ = 1). Pearson’s correlation coefficients were negative (r = −0.035) between linkage disequilibrium (R^2^) and physical distance (bp), as well as between *p*-value and R^2^ (r = −0.40), showing the existence of linkage decay. Linkage disequilibrium (LD) decay differed across the chromosomes, ranging from 8289 bp for chromosome 1 to 58,562 bp for chromosome 11.

### 2.3. Genome-Wide Association Scan for Tuber Dry Matter Content

We identified three SNP markers significantly associated with DMC (Figure 2, Table 3). Of these three SNP loci, two (Chr6_59775 and Chr6_615325) were on chromosome 6 at 59,775 and 615,325 bp physical positions, respectively (Table 3). These two SNP loci had marker effects of −4.01 and −3.33, respectively, and explained 15.50% of the total phenotypic variation. The third SNP locus associated with the tuber DMC was on chromosome 19 (Chr19_8692) at 8692 bp physical position. This SNP had a marker effect of 1.39 and explained 30.37% of the total phenotypic variation on DMC (Table 3). Of the four-gene action models (general, additive, simple dominant (1-dom-alt) and dominant reference (1-dom-ref) used in our analysis, one SNP (Chr19_8692) was identified through the additive gene action model, while two SNPs (Chr6_615325 and Chr19_8692) were identified through the dominant reference model (Figure 2).

The three QTLs associated with DMC showed significant QTL × location interaction (Table 4). The SNP marker Chr19_8692 was significant at each of the three locations, while the two markers on chromosome 6 were significant in two (Ibadan and Ikenne) of the three locations (Table 5). The Quantile–Quantile (QQ) plot corroborated with reducing −log10 (*p*-value) toward the expected level for the dry matter (Figure 2). Further dissection of the two SNP loci associated with the tuber DMC on chromosome 6 showed that clones with the homozygous allele AA possessed higher tuber DMC than those with the heterozygous allele AG and/or homozygous allele GG (Figure 3). A marker effect on chromosome 19 revealed the allele CC to be linked with low DMC in the studied population, while allele CT and TT accounted for high DMC (Figure 4).

### 2.4. Genome-Wide Scan for Oxidative Browning

We identified two SNP markers, “Chr5_118279” and “Chr5_125093”, both on chromosome 5 that had a significant association with the oxidative browning (Figure 5). The two significant SNP markers, “Chr5_118279” and “Chr5_125093”, with marker effects of −4.06 and −3.48, were detected at LOD scores of 4.30 and 4.19, respectively (Table 3). Three (general, additive, and dominant alternative) of the four gene action models used in the analysis showed a significant marker–trait association for OxB on chromosome 5 that explained 12.74% of the total phenotypic variation (Table 3). The genotypic status of significant QTL markers associated with OxB and their corresponding mean phenotypic values presented in Figure 6. The two SNP markers were significant at Ikenne and Ibadan but not at Ubiaja (Table 5). The average phenotypic values of tuber oxidation for GA and GG genotypes were 5.22 and 7.36 for Chr5_11279, 4.91 and 7.48 for Chr5_125093 (Figure 6). Analysis of variance (ANOVA) revealed that the OxB value for the allele GA was significantly lower than that predicted by the allele GG (Figure 6).

### 2.5. Gene Annotations

The significant SNP loci were related to structural genes using a candidate approach that takes advantage of the protein-coding gene annotation of the *D. alata* genome in preparation [20] (used with permission). This approach resulted in the identification of candidate genes on chromosomes 6 and 19 associated with DMC and chromosome 5 for the OxB. Through the annotation, various putative genes (26) associated with DMC, including ATP-grasp fold and succinyl-CoA synthetase-type (IPR013650) were identified at 4 kb from Chr19_8692, while glycoside hydrolase family 9 (IPR001701) and six-hairpin glycosidase superfamily (IPR008928) were identified on the marker associated with DMC on chromosome 6. For OxB, six putative candidate genes, including the thioesterase domain (IPR006683), phenylacetic acid degradation-related domain (IPR003736), histone deacetylase family (IPR000286), histone deacetylase domain (IPR023801), DUF630 (IPR006868), Cwf19-like and C-terminal domain-1 (IPR006768) were identified near the peak SNPs.

LD block heatmaps based on the LD of each identified SNP loci are shown in Figure 7 and Figure 8. For DMC, the LD analysis of the three loci (two on chromosome 6 and one on chromosome 19) showed that these markers had a relatively average to high LD parameter (R^2^  > 0.8), showing a relatively high correlation (Figure 7). For OxB, the LD analysis on two loci (two markers on chromosome 5) showed that these markers had relatively low LD parameters (R^2^  < 0.6) (Figure 8).

## 3. Discussion

Dry matter content and oxidation properties of yam tubers are very important quality traits that influence the rate of adoption of new clones for cultivation and consumption. Yam improvement efforts worldwide, especially in West Africa, have tested several clones within yam populations for DMC and none or minimal oxidation of fresh tuber using conventional selective breeding based on phenotypic records [8,22,23], an approach that is slow and arduous. For quality traits, DNA-based strategies such as GWAS reported in this study have advantages over the conventional selection breeding approach because it has the potential to fast-track the development and delivery of improved yam varieties with acceptable end-user attributes. The potential of GWAS to dissect complex traits has been proven in root and tuber crops such as cassava [24,25,26] and potatoes [27,28]. The present GWAS sought to identify QTL (s) and putative candidate genes associated with genetic variation in DMC and OxB in water yam. The highly significant genotype variance for DMC content and OxB in the current *D. alata* panel warrant further analysis for the dissection of the genetic basis of variation for these two traits.

Detailed knowledge of population structure and familial relationships (kinship) in the association panel is crucial to prevent sham associations in GWAS [29]. Population structure and admixture for this population were reported in a previous study [21]. The Q matrix (population structure) and K matrix (Admixture) were used as covariates in a mixed linear model for the association analysis to reduce false-positive associations. The reducing −log10 (*p*-values) toward the expected level for both traits on the quantile–quantile plots is a sign that the model successfully accounted for population structure and familial relationships in the GWAS analysis.

Three QTLs were identified to be associated with DMC, 2 of which showed significant QTL-by-Environment interactions (QEI). In a study to identify QTLs associated with cassava brown streak disease, Kayondo et al. [30] reported similar results with QTLs identified at different locations and highlighted the effects of many factors such as the panel size, harvest time and environmental conditions influencing QTL identification. To address this, the best approach for increasing the resolution of associations of traits and QTLs is to combine multi-location genotypic and phenotypic scores from different diversity panels [30,31]

Using the marker effect, we observed allele AA on chromosome 6 and allele TT on chromosome 19 to be responsible for high DMC in the diversity panel used in the study. Information on marker effect through the segregation pattern is fundamental for marker validation and deployment in a breeding programme [32,33,34]. We also identified the heterozygous allele GA to be significantly associated with low OxB.

Our study also identified putative candidate genes within the QTL regions of the targeted traits. A total of 26 putative candidate genes were detected upstream and downstream of the SNP associated with DMC, of which 3 genes (Serine/threonine-protein kinase, Tetratricopeptide-like helical domain superfamily and Glycoside hydrolase family 9) were reported to play important roles in DMC. Serine/threonine-protein kinase (SnRK1) was reported to participate in the process of starch and sugar biosynthesis in potatoes and stimulated glucose pyrophosphorylase [35,36,37]. In potato (*Solanum tuberosum*) and wheat (*Triticum aestivum*), SnRK1 was reported to stimulate some enzymes in the starch biosynthesis pathways [38,39]. The Tetratricopeptide-like helical domain superfamily genes were reported to mediate protein–protein interactions and involved in the production of protein and starch that are the principal storage carbohydrates in plants [40,41]. The third important putative gene “Glycoside hydrolase family 9” was reported to be involved in diverse enzymatic metabolisms of carbohydrate compounds available in many plant tissues [42].

Six putative candidate genes were identified within the QTL regions of the peak SNPs detected for OxB. Of these 6 genes, the Thioesterase domain (IPR006683) has been reported to play a major role in tuber oxidative browning pathway [43]. The thioesterase domain is part of peroxisomes that contain soluble thioesterases (oxalate oxidase), which play a significant role in regulating flux of various substrates by releasing CoA from β-oxidation intermediates and products [44]. The oxalate oxidase is an enzyme containing manganese that catalyse the oxidation of oxalate to carbon dioxide by reducing the oxygen to hydrogen peroxide [45].

Tuber dry matter had a negative correlation with OxB. This desirable correlation suggests that selection for high DMC is expected to reduce enzymatic browning following oxidation of the tuber flesh. The genotypic factors greatly contribute for trait linkages and the negative correlations between the two phenotypic traits may be an indication of pleiotropism, genetic coupling and/or linkage disequilibrium with population structure effects [46]. However, our GWAS result did not reveal co-localized SNP markers for both traits to confirm that such an association is due to pleiotropy. In any case, the observed trait association provided the opportunity to select superior yam clones that indeed offer increased tuber dry matter and less tuber flesh enzymatic oxidation simultaneously. Our study identified significant and functional SNPs established at the genome level and revealed the known genes, which showed the presence of the disequilibrium between SNP markers and causative variants of DMC and OxB within or near identified genes. It is plausible that the allelic variation for oxidation observed in our study, which is associated with browning, is the result of enzymatic activities in yam, which includes polyphenol oxidase and peroxidase activities that play a major role in the phenolic content of yam tubers [47,48].

The functions and characteristics of some identified genes have not been explored. However, the identified genes from this study may provide new intuition into the genetic fundamentals of DMC and OxB in *D. alata*.

## 4. Materials and Methods

### 4.1. Plant Materials

A total of 100 *D. alata* clones, made up of breeding lines and landraces, obtained from the Yam Breeding Unit of the International Institute of Tropical Agriculture (IITA), Ibadan, Nigeria, were utilized for this study. The clones were planted at three locations in Nigeria in the 2018 growing season: Ibadan (72°4′ N, 3°54′ E), Ubiaja (6°39′ N, 6°22′ E) and Ikenne (6°58′ N, 4°0′ E). A lattice design with two replications was used in each of the three locations.

### 4.2. Phenotypic Data Collection

#### 4.2.1. Tuber Dry Matter Content

Healthy yam tubers were sampled in each replication for dry matter determination. After harvest, the fresh tubers of each clone were cleaned with water to remove soil particles. Thereafter, the tubers were peeled and sliced into small sizes for easy oven drying; 100 g of freshly grated tuber flesh sample was weighed, put into a Kraft paper bag and dried at 105 °C for 16 h. After drying to s constant weight, the weight of each samples was recorded, and the DMC was determined using the following formula:(1)$$\%\;Dry\;matter\;content\left(DMC\right)=\frac{weight\;of\;dry\;sample\left(g\right)}{\;weight\;of\;wet\;sample\left(g\right)}\times 100$$

#### 4.2.2. Oxidative Browning

After harvest, one well-developed and mature representative tuber was sampled in each replication. The sampled tuber was peeled, cut and chipped with a hand chipper to get small thickness size pieces (5 cm and 0.5 mm thickness). A chromameter (CR-410, Konica Minolta, Japan) was used to read the total colour of sampled pieces placed on a petri dish immediately after tubers cut and exposure to air (0 min) and 30 min after. The lightness (L*), red/green coordinate (a*) and yellow/blue coordinate (b*) parameters were recorded for each chroma meter reading for the determination of the total colour difference. A reference white porcelain tile was used to calibrate the chromameter before each determination [49].

The total colour difference (Δ*E**) between all the three coordinates was determined using the following formula [50]:(2)$$\Delta E^{*}={\left[\Delta L^{*2}+\Delta a^{*2}+\Delta b^{*2}\right]}^{1/2}$$
where Δ*E** = the total colour difference, Δ*L** = the difference in lightness and darkness (+ = lighter, − = darker, Δ*a** = the difference in red and green (+ = redder, − = greener) and Δ*b** = the difference in yellow and blue (+ = yellower, − = bluer).

Oxidative browning was estimated from the total variation from the difference in the final and initial colour reading as:(3)Oxidative browning (OxB)=ΔEF−ΔEI
where Δ*EF* = the colour reader value at final time (30 min) and Δ*EI* = the initial colour reader value at 0 min.

### 4.3. DNA Extraction, Library Construction and SNP Calling

About one gram of fresh, healthy and young leaves was collected from a field-grown plant of each clone and placed on dry ice immediately. The leaf samples were lyophilized and kept at under room temperature. DNA was extracted from lyophilized leaf samples using the CTAB (cetyltrimethylammonium bromide) protocol [51] with slight modification. The DNA quality was assessed on 0.8% agarose gel and concentration was estimated using nanodrop (Amersham Bioscience, Piscataway, NJ, USA) following the manufacturer’s instructions. Subsequently, 50 µL of 50 ng/µL diluted DNA of each clone was prepared and sent to Diversity Arrays Technology (DArT) Pty Ltd., Australia for a genome scan using the DArT marker procedure described by Agre et al. [21].

#### 4.3.1. Phenotypic Data Analysis

Phenotypic data obtained from the three locations were pooled and subjected to a linear mixed model analysis using the lme4 package implemented in R [52]. The best linear unbiased estimates (BLUEs) for three locations were obtained by considering clone main effect as fixed and location and replication effect as random in the mixed model as follows:(4)$$Y_{ijkl}=\mu +B{\left(E\right)}_{j\left(i\right)}+G_{k}+GE_{ij}+e_{ijkl}$$
where *Y_ijkl_* = phenotypic observation for a trait, *µ* = grand mean, *E* = environment effect (location), *B*(*E*) = replication effect nested in location, *G* = genotype effect, *GE* = genotype by environment interaction, *e* = random residual error.

#### 4.3.2. Genotypic Data Analysis

Multiple sequences were generated by the DArTSeq platform using proprietary analytical pipelines (Diversity Array Technology, Canberra, Australia) and mapped to the *Dioscorea alata v2* reference genome assembly in preparation [20] (used with permission) using a local BLAST server. This produced a raw dataset of 22,140 SNPs that were subjected to quality control filtering with the following criteria: markers with low sequence depth <5; SNP markers with missing values >20%; minor allele frequency (MAF) <0.05; genotype quality <20; and heterozygosity >50; which resulted in 9687 good-quality SNPs. These filtered 9687 SNPs were subjected to summary statistics such as minor allele frequency, polymorphism information content (PIC), observed and expected heterozygosity (OH/EH) using PLINK 2 [53]. The SNP distributions and density across linkage groups obtained by using bin.size1e6 for scaling the physical position from bp to Mb in the CMplot package [54].

#### 4.3.3. Linkage Decay, GWAS Analysis and Gene Annotation

A mixed linear model implemented in the GWASpoly package in R was used to compute associations using the mixed model y=Xb+Zu+e [55], where y is the vector of the phenotypic observations (BLUEs for DMC and OxB), *X* represents the SNP markers (fixed effect), *Z* represents the random kinship (co-ancestry) matrix, *b* is a vector representing the estimated SNP effects, *u* is a vector representing random additive genetic effects, and *e* is the vector for random residual errors. A co-ancestry matrix from ADMIXTURE and kinship were included as covariates in the GWASpoly to account for population structure and familial relatedness, respectively, to reduce spurious associations. Four different gene action models (additive, general, dominant alternative (1-dom-alt) and dominant reference (1-dom-ref)) were considered for the trait association study [56]. The GWAS analysis was performed using BLUEs across locations and significant QTLs were determined based on the *p*-value *(p* < 0.05*)* adjusted by the Bonferroni correction implemented in GwasPoly [56].

Quantile–quantile (QQ) plots were generated by plotting the negative logarithms (−log_10_) of the *p*-values against their expected *p*-values to fit the appropriateness of the GWAS model with the null hypothesis of no association and to determine how well the models accounted for population structure. The Manhattan plot was created for visualizing GWAS on the entire genome and zoom mapping was performed on a particular chromosome after identifying a significant SNP marker.

The marker effect or SNP contribution was estimated for the significant SNPs using a multiple regression analysis using *lm* function implemented in R where the trait was considered as a response variable while the SNP markers above the Bonferroni threshold for the trait the independent variable.

For gene annotation, linkage disequilibrium (LD) was evaluated between the significant SNPs at 5 kb (downstream and upstream) using the LDheatmap library [57]; the generic feature format (GFF3) of the reference genome was used to identify the main gene in the inter-genic region using the SNPReff. Public database Interpro, European Molecular Biology Laboratory-European Bioinformatics Institute (EMBL-EBI) [58] was used to determine the functions of the genes associated with the different SNPs identified.

## 5. Conclusions

Tuber dry matter and oxidative browning are key quality attributes that determine cultivar acceptability in water yam. The results of this study represent significant progress toward dissecting the genetic architecture of these two key traits in yam breeding. We reported three SNP markers associated with DMC and two SNP markers for OxB. Markers associated with these key traits showed significant phenotypic variation, which could be useful to breeders for marker-assisted selection (MAS) in the yam breeding program. This study provides the basis for a long-term collective exploration to determine valuable genes and alleles from the yam and consequently yam varieties’ improvement.

## Figures and Tables

**Figure 1 plants-09-00969-f001:**
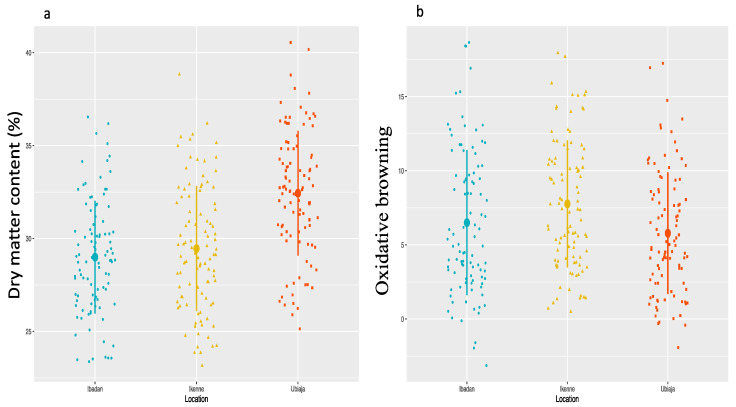
Dot plots for the distributions of tuber dry matter content (**a**) and oxidative browning (**b**) across three locations. The dots represent the clones, while the solid line represents the error bar.

**Figure 2 plants-09-00969-f002:**
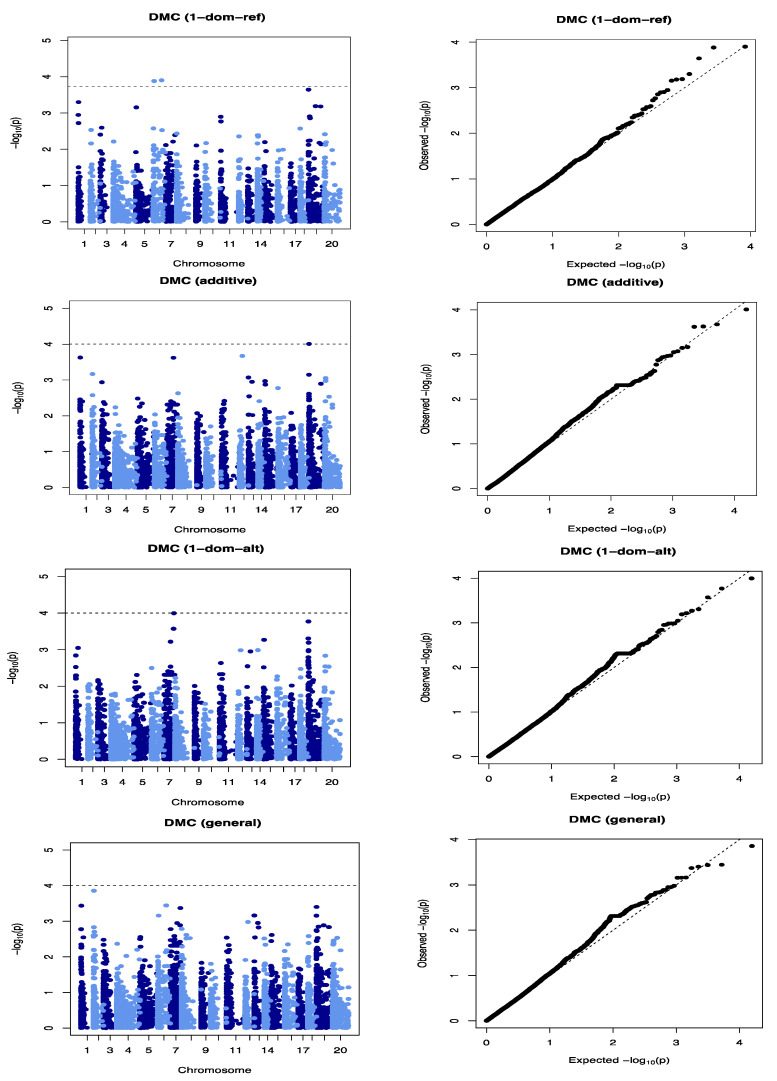
Genome-wide association analysis of tuber DMC. Manhattan and Quantile–Quantile plots for DMC on chromosomes 6 and 19. The dashed lines on the Manhattan plot represent the significant threshold.

**Figure 3 plants-09-00969-f003:**
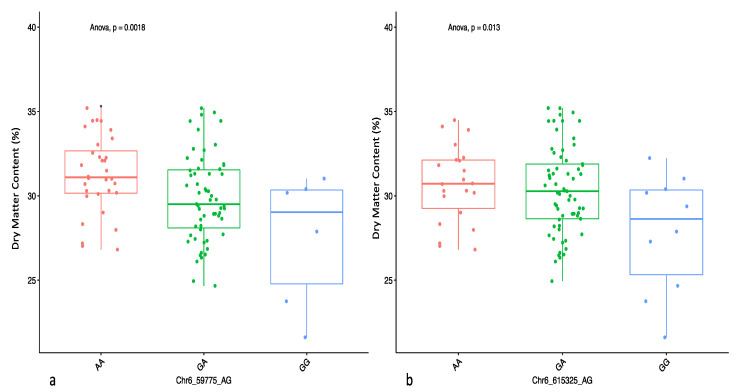
Boxplots showing the effect of the significant markers associated with tuber DMC on chromosome 6 with two SNPs: (**a**) Chr6_59775 and (**b**) Chr6_615325. The letters on the X axis represent allele variants (AA, GA, and GG); significant codes: * = 0.05 and *p* represents the analysis of variance probability value associated with the variation across variants.

**Figure 4 plants-09-00969-f004:**
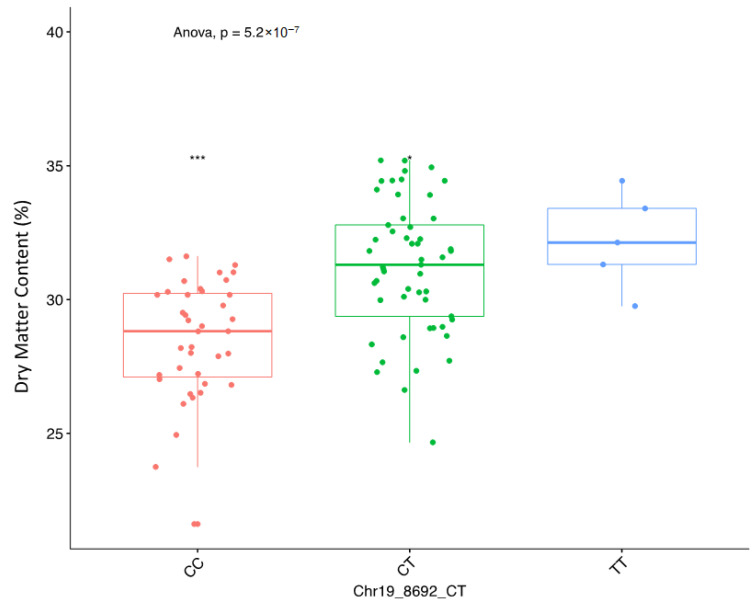
Boxplots showing the effect of the significant markers associated with tuber DMC on chromosome 19. The letters on the X-axis represent allele variants (CC, CT, and TT); *p* represents the analysis of variance probability value associated with the variation across variants.

**Figure 5 plants-09-00969-f005:**
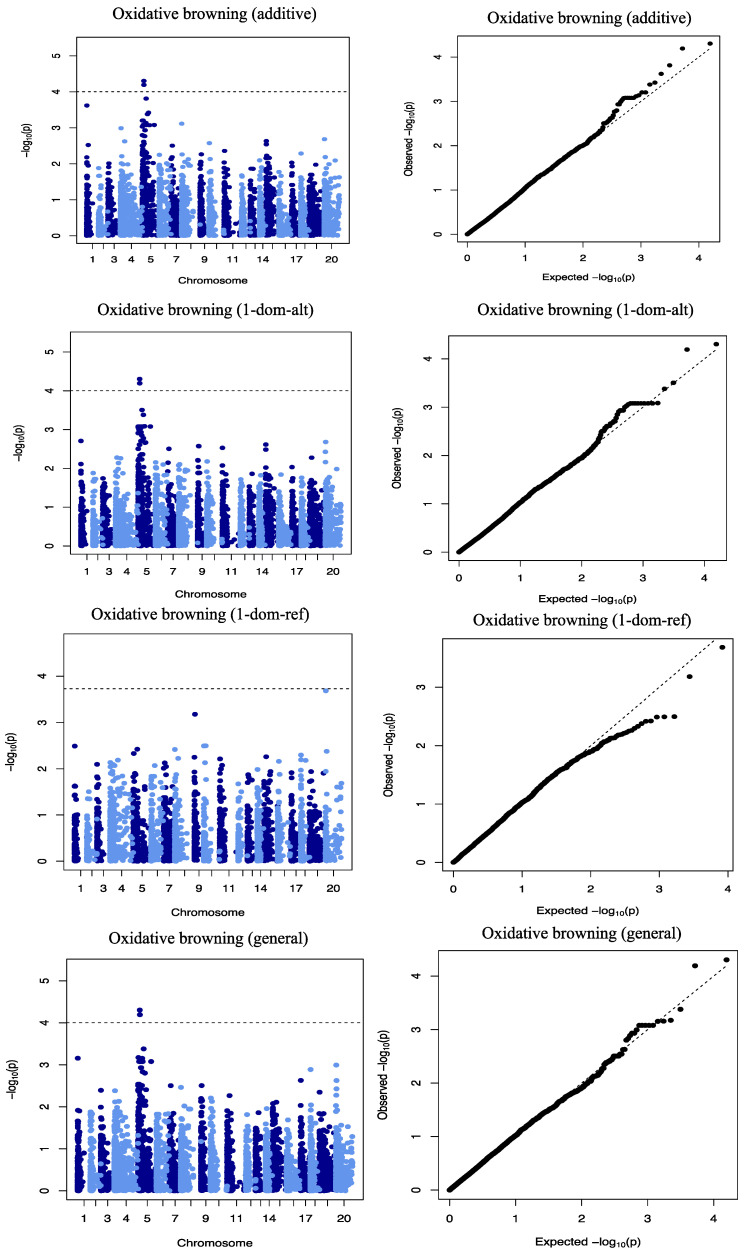
Genome-wide association analysis for OxB. Manhattan and Quantile–quantile plots for OxB on chromosome 5. The dashed lines on the Manhattan plot represent a significant threshold.

**Figure 6 plants-09-00969-f006:**
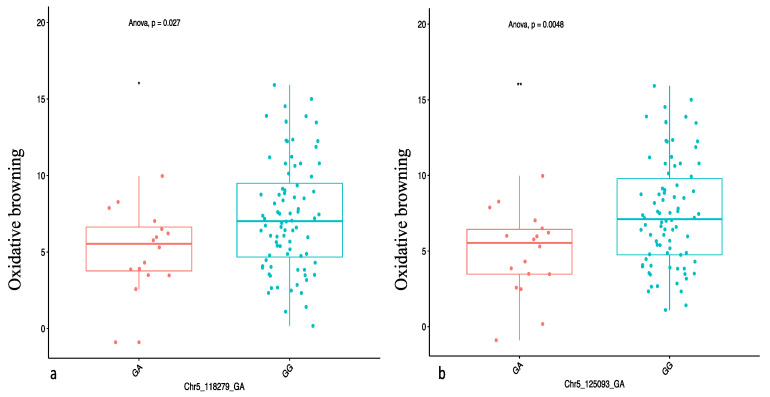
Marker effect of the significant markers associated with OxB on chromosome 5 with two markers: (**a**) chr5_118279 and (**b**) chr5_125093. The letters on the X-axis represent allele variants (GA and GG). Significant codes: ** = 0.01; * = 0.05 and *p* represents the ANOVA probability value associated with the variation across variants.

**Figure 7 plants-09-00969-f007:**
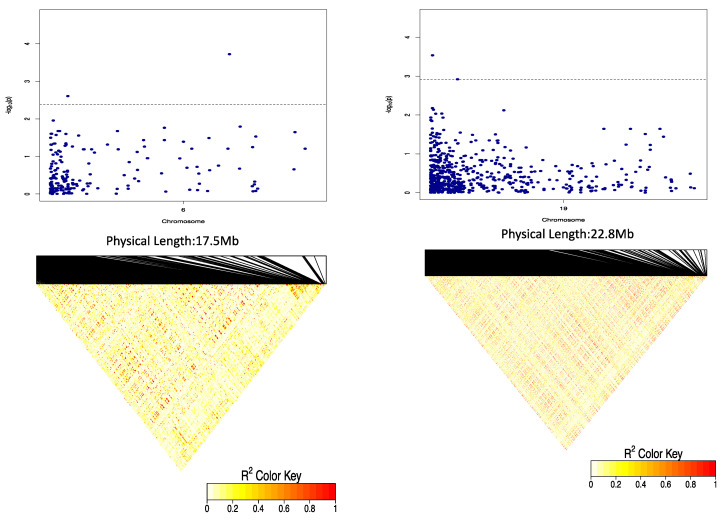
Linkage disequilibrium (LD heatmap) showing pairwise LD between the SNP markers covering the entire chromosomes 6 and 19, respectively, carrying the genes encoding for tuber DMC. The red colour shows the markers with high LD followed by yellow then the white colour. The dashed lines on the Manhattan plot represent the significant threshold while the *X*-axis presents physical distances.

**Figure 8 plants-09-00969-f008:**
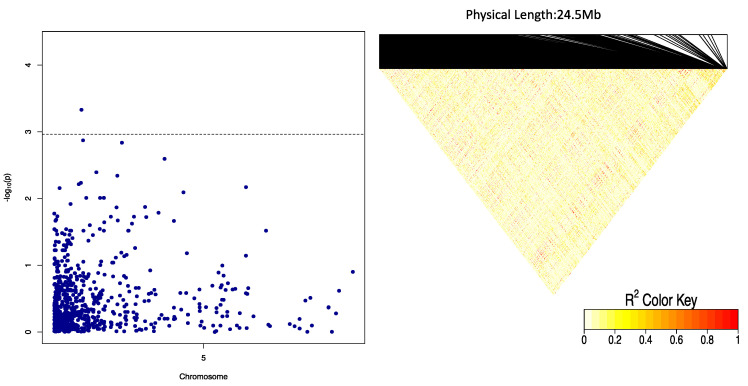
Linkage disequilibrium (LD) heatmap showing pairwise LD between the SNP markers covering the entire chromosome 5 carrying genes encoding for OxB. The red colour shows the markers with high LD followed by yellow then the white colour. The dashed lines on the Manhattan plot represent the significant threshold while the X-axis presents physical distances.

**Table 1 plants-09-00969-t001:** Estimates of variance for total genotypic and genotype × location effects in a panel of 100 water yam clones phenotyped at three locations.

Source of Variance	Dry Matter Content		Oxidative Browning	
	Variance	Standard Deviation	Variance	Standard Deviation
**Genotype**	6.315	2.512901	47.8805	6.9196
**Location**	3.92	1.980232	7.8743	2.8061
**Replication**	7.294 × 10^−2^	0.270077	7.8743	2.8061
**Genotype × location**	1.806	1.343863	20.9957	4.5821
**Residual**	6.630	2.574859	75.4086	8.6838

**Table 2 plants-09-00969-t002:** Number of single nucleotide polymorphism (SNP) *D. alata* chromosomes before and after filtering alongside the average polymorphism information content (PIC).

Chromosome	All SNPs	Filtered SNPs	Chr Size (Mb)	PIC of Filtered SNPs
1	698	299	18.5	0.218
2	898	335	14.5	0.224
3	635	432	19.6	0.219
4	1225	893	33.4	0.241
5	1143	923	26.3	0.223
6	442	360	16.2	0.276
7	735	583	16.5	0.265
8	943	640	28.4	0.217
9	996	346	14	0.221
10	1096	326	23.6	0.277
11	1186	392	26.7	0.262
12	765	329	14	0.225
13	839	304	15	0.226
14	1080	528	11.2	0.237
15	583	424	20.1	0.255
16	944	386	19.3	0.243
17	1136	436	14.2	0.230
18	1113	477	12.2	0.208
19	1046	843	23.7	0.213
20	564	431	23.1	0.229
Total	18,067	9687	390.5	

**Table 3 plants-09-00969-t003:** Summary of significant single nucleotide polymorphism (SNPs) describing different genomic regions associated with tuber DMC and OxB in a panel of 100 *D. alata* clones.

Traits	SNP Markers	Chr	Alleles	Model	Physical Position (bp)	LOD	Effect	R^2^%
DMC	Chr19_8692	19	CT	additive	8692	4.01	1.79	30.37
	Chr6_59775	6	GA	1-dom-ref	59,775	3.88	−4.01	8.06
	Chr6_615325	6	GA	1-dom-ref	615,325	3.90	−3.33	7.44
OxB	Chr5_118279	5	GA	Additive, general, 1-dom-alt	118,279	4.30	−4.06	4.91
	Chr5_125093	5	GA	Additive, general 1-dom-alt	125,093	4.19	−3.48	7.83

Chr: chromosome, Pos: position; LOD: logarithm of the odds; R^2^ = r-square.

**Table 4 plants-09-00969-t004:** Markers associated with the tuber DMC and OxB with their variation across locations.

Traits	Marker Interaction	MS	*p*-Value	Adjusted R
DMC	Chr6_59775: Location	124.68	8.822 × 10^−14^ ***	0.2139
	Chr6_615325: Location	122.112	2.081 × 10^−13^ ***	0.209
	Chr19_8692: Location	191.324	2.2 × 10^−16^ ***	0.343
OxB	Chr5_118279: Location	67.829	0.01475 *	0.03051
	Chr5_125093: Location	90.377	0.001929 **	0.0463

MS: Mean square. *p* represents the analysis of variance probability value associated with the variation across variants.

**Table 5 plants-09-00969-t005:** QTL interaction across each location.

	Marker Location	*p*-Value
DMC	Chr6_59775: Ikenne	0.00532 **
	Chr6_59775: Ibadan	0.00762 **
	Chr6_59775: Ubiaja	0.21641
	Chr6_615325: Ikenne	0.003599 **
	Chr6_615325: Ibadan	0.000982 ***
	Chr6_615325: Ubiaja	0.467
	Chr19_8692: Ikenne	0.000207 ***
	Chr19_8692: Ibadan	2.83 × 10^−^^5^ ***
	Chr19_8692: Ubiaja	0.000617 ***
OxB	Chr5_118279: Ikenne	0.00699 **
	Chr5_118279: Ibadan	0.02513 *
	Chr5_118279: Ubiaja	0.19550
	Chr5_125093: Ikenne	0.002499 **
	Chr5_125093: Ibadan	0.012047 *
	Chr5_125093: Ubiaja	0.146802

*p* represents the analysis of variance probability value associated with the variation across variants.

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
