# Peer review of "Genome-Wide Association Analysis for Tuber Dry Matter and Oxidative Browning in Water Yam (Dioscorea alata L.)"

_plants, 2020, doi:10.3390/plants9080969_

Round 1
Reviewer 1 Report
The manuscript has been improved. However, the authors might consider abreviating oxidative browning (as 'OxB' or similar) throughout the MS. Also, be consistant in the use DMC (see line 157). Line 232 defines DMC again - not necessary.
Author Response
Dear Reviewer, We do really appreciate your comment and suggestion toward the improvement of the manuscript.
We have used the term "OxB" for the oxidation browning across the document and we have been consistent by using the "DMC" in the MS.
Reviewer 2 Report
I still do not understand why SNPs are very densely distributed on one end of each of all of the 20 chromosomes (in distal ca. 2 Mb sections) and than they are gradually getting more sparse towards the other end (Figure 2). It seems like a technical artifact. Gaps are not a problem, it is natural that there are regions more or less saturated with SNPs, the problem is the gradient of SNP densities across chromosomes.
Also, Figures 8 and 9 should be further edited. Please, add Mb scales to X axes and explain it the LD graphs show entire chromosomes or just regions identified as associated with DMC (fig. 8) or oxidative browning (fig. 9)
Author Response
We appreciate the point raised by the reviewer here.
Figure 2 has been corrected and cross-checked with the genome reference.
The new correction on figure 2 addressed the SNP density across the chromosomes.
The fig 8 and 9 have also been corrected and the new LD was done on the entire chromosome respectively.
Reviewer 3 Report
I am not comments
I agree with the corrections
Author Response
Thanks for reviewing our manuscript.
This manuscript is a resubmission of an earlier submission. The following is a list of the peer review reports and author responses from that submission.
Round 1
Reviewer 1 Report
My only concernt was the use of Blast to align short sequences to the yam reference genome. I think it would have been more appropriate to use a tool like bwa or bowtie2 which were specifically designed for such purpose. Nevertheless, blast results are still reliable enough to support the results and conclussions in this study.
Reviewer 2 Report
General Comments:
Appears to be well done and clearly presented.
Some minor grammatical errors
Line 19 chromosomes
45 types of food preparation,
Figure 2 (top). Title font is off
132 marker effects
134 Use DMC for dry matter content (be consistent throughout the MS)
144 as previous
160 allele variants
244 same as 134
249 conditions
256 allele?
257-59 Seems out of place. Move elsewhere?
261 upstream and downstream
305 to obtain thin slices (can you state how thick they were?)
306 Petri dish
324 room temperature
425 Check References to italicize scientific names
483 same as previous
535 Plant Molec. Biol. Reptr.
Reviewer 3 Report
The authors present results on the identification of QTLs for dry matter content and oxidative browning in Discorea alata based on a multi-environment experiment comprising 100 D. alata clones. While potentially interesting, the study relies heavily on the unpublished data on the assembly of the D. alata genome. I suggest not to consider the manuscript for publication until the genome assembly paper is released. It is crucial to present the genome information in the Introduction of the present manuscript, including the genome size, number of genes, etc. Subsequently, when discussing results, the authors indicate a handful of candidate genes, but obviously no precise information on these genes can be given except of their putative function. Positions of SNPs are also of no value to the reader as long as the assembly is not available. For future reference, when preparing the resubmission: There are bits of information that seem confusing. On Figure 2, D. alata chromosomes are presented and the largest chromosome (Chr 4) is only 2 Mb-long, implying that the size of whole genome is only ca. 30 Mb. Also, most SNPs are located in one end of all chromosomes and then their density is gradually decreasing with large portion of chromosomes completely lacking any SNPs. The apparently non-random distribution requires a comment. How does it affect GWAS (considering possible QTLs in regions not covered by SNPs)? The scale above is 0 Mb (3x), 1 Mb (5x) and 2 Mb - I am not sure what it refers to. On the other hand, positions of QTL-associated SNPs are given in kbs, e.g 615325kb (line 131), which equals to 615 Mb, largely exceeding the size of one chromosome. No Bonferroni-corrected -log10(p-values) are given. The dashed lines on Figures 3 and 6 (presumably reflecting the sigificance threshold, it has not been explained) point at the -log10(p) of 4 (except for the topmost graph on Figure 3, where it is lower - why?). On Figures 8 and 9 dashed lines are between 2 and 3 (why?) and no information is given about the x-axis (supposedly, it should present physical distances). Minor: The term 'intergenic region' is not used properly (e.g. legends of Figures 8 and 9: "...intergenic region [...] carrying genes encoding...") The term "tuber flesh oxidation" should be replaced by the commonly accepted "oxidative browning"
Reviewer 4 Report
This paper is an interesting contribution to better understand yam genetics and opens up many possibilities for the world of genetic improvement in this species. I think it is well written. Note that I am not an expert in statistics and therefore I cannot evaluate the statistical analyzes carried out. It seems to me that they have consistent results and that the discussion is adequate and clarifies the previously stated results. The conclusions are also concise and decisive. Introduction have old references
I have found some references not included in the text that it would improve the introduction and the discussion:
Pétro, D., Onyeka, T. J., Etienne, S., & Rubens, S. (2011). An intraspecific genetic map of water yam (Dioscorea alata L.) based on AFLP markers and QTL analysis for anthracnose resistance. Euphytica, 179(3), 405-416.
Siqueira, M. V., Bonatelli, M. L., Günther, T., Gawenda, I., Schmid, K. J., Pavinato, V. A., & Veasey, E. A. (2014). Water yam (Dioscorea alata L.) diversity pattern in Brazil: an analysis with SSR and morphological markers. Genetic resources and crop evolution, 61(3), 611-624.
Sartie, A., Asiedu, R., & Franco, J. (2012). Genetic and phenotypic diversity in a germplasm working collection of cultivated tropical yams (Dioscorea spp.). Genetic resources and crop evolution, 59(8), 1753-1765.
